# Factors Influencing Exclusive Breastfeeding During the Postpartum Period: A Mixed-Methods Study

**DOI:** 10.3390/nu17182992

**Published:** 2025-09-18

**Authors:** Greyce Minarini, Eliane Lima, Karla Figueiredo, Ana Paula Carmona, Mariana Bueno, Nátaly Monroy, Cândida Primo

**Affiliations:** 1Center for Health Sciences, Maruípe Campus, Federal University of Espírito Santo, Avenida Marechal Campos, 1.468, Vitória 29047-105, ES, Brazil; greyce.minarini@gmail.com (G.M.);; 2Nursing Department, Federal University of Paraná, Av. Prefeito Lothário Meissner, 632, Bloco Didático II, Jardim Botânico, Curitiba 80210-170, PR, Brazil; 3Nursing Research, Innovation, and Development Centre of Lisbon—[CIDNUR], Escola Superior de Enfermagem de Lisboa, Avenida Professor Egas Moniz, 1600-190 Lisboa, Portugal; anapcarmona@esel.pt; 4Lawrence Bloomberg Faculty of Nursing, University of Toronto, 155 College Street, Suite 130, Toronto, ON M5T 1P8, Canada; 5Department of Statistics, Goiabeiras Campus, Federal University of Espírito Santo, Av. Fernando Ferrari, 514—Goiabeiras Campus, Vitória 29075-910, ES, Brazil

**Keywords:** breastfeeding, prenatal education, labor and delivery, postpartum period

## Abstract

**Background/Objectives:** Breastfeeding is essential to maternal and child health, and multiple factors influence its success. This study examined the factors associated with breastfeeding type among infants aged 0 to 12 weeks. **Methods:** A mixed-methods study, employing a convergent design, was conducted in the rooming in unit of a hospital in Espírito Santo, Brazil. A total of 296 mothers of neonates ≥ 34 weeks participated in both the quantitative and qualitative phases. The qualitative phase involved semi-structured interviews conducted in the hospital setting. In the quantitative phase, data were collected via telephone in three waves (on days 14, 40, and 90 postpartum), critical moments for establishing and maintaining breastfeeding, analyzing sociodemographic factors (age, education, marital status, number of pregnancies), clinical factors (gestational age, mode of delivery, milk production) and support factors (social and hospital). Descriptive statistical analysis and binomial and multinomial logistic regression models were used, conducted in R 4.3.3 software. The qualitative and quantitative findings were integrated through simultaneous incorporation and presented in a joint display. **Results:** The analysis showed that although most mothers had high adherence to prenatal care, breastfeeding counseling was insufficient. In addition to the type of delivery and immediate skin-to-skin contact, other factors were also found to be relevant to maintaining exclusive breastfeeding. Higher maternal education and a greater number of pregnancies were associated with better breastfeeding practices, albeit with variations in statistical significance. Support received during hospitalization, especially from the healthcare team, also emerged as a central element in the qualitative reports, reinforcing its role as a protective factor for continued breastfeeding. Early formula use within the first 48 h was identified as a barrier to initiating and maintaining breastfeeding. **Conclusions:** The duration and maintenance of exclusive breastfeeding varied over time, depending on factors such as the number of prenatal appointments, education level, number of pregnancies, mode of delivery, immediate skin-to-skin contact, and, most importantly, the use of formula in the first 48 h. The early introduction of formula in maternity wards represented a significant obstacle to breastfeeding, reinforcing the importance of integrated public policies and multidisciplinary initiatives that promote breastfeeding from birth.

## 1. Introduction

Breastfeeding is widely recognized as the optimal form of nutrition for neonates, offering significant health benefits for both mothers and infants. The World Health Organization (WHO) recommends exclusive breastfeeding for the first six months of life and, subsequently, supplementing with other foods until two years of age or more. However, the type of food (e.g., exclusive, mixed, or formula predominant) and duration vary across populations and are shaped by a complex interplay of sociodemographic, clinical, and behavioral factors. Education provided during the prenatal period, labor and delivery, and the postpartum period plays a critical role in this process by influencing mothers’ decisions to effectively initiate and sustain breastfeeding [1,2,3].

Studies have shown that the relationship between perinatal counseling and breastfeeding duration is mediated by multiple factors, including individual characteristics, infant care practices, social support, and health policies [1,4]. While previous research has examined these determinants, a deeper understanding of how they interact in specific contexts is still needed to inform more effective public health policies, healthcare, and health interventions.

Although comprehensive global studies on exclusive breastfeeding exist, few specifically investigate the factors that influence this outcome in Brazilian populations. Studies [5,6] provide relevant contributions, but do not consider the cultural, social, and public health particularities of Brazil. Our study seeks to fill this gap by analyzing a representative sample of Brazilian mothers, enabling a greater understanding of the local factors that impact exclusive breastfeeding. Furthermore, it is distinguished by its use of mixed methods in a longitudinal approach, following 274 women for three months. This analytical strategy allows us to capture changes over time and generates more dynamic and robust evidence than that obtained from cross-sectional studies.

Considering the multitude of factors that influence breastfeeding and breastfeeding outcomes, adopting a comprehensive approach that incorporates both quantitative and qualitative data is essential to explore this complex phenomenon. Mixed-methods research (MMR) offers a valuable strategy by combining quantitative data (QUAN) to statistically assess associations between sociodemographic and clinical variables and qualitative data (QUAL) to explore the subjective and behavioral dimensions of breastfeeding, including the influence of counseling received throughout the perinatal period [7,8,9,10]. This integrated approach enables a more thorough exploration of complex phenomena than either method alone [10].

Based on these considerations, this study hypothesized that breastfeeding counseling received during the prenatal period, labor and delivery, and the postpartum period positively influences breastfeeding type and duration. This study examined the factors associated with breastfeeding type among infants aged 0 to 12 weeks.

## 2. Materials and Methods

### 2.1. Study Design

This was a mixed-methods study employing a convergent design, integrating quantitative and qualitative components to provide a more comprehensive and in-depth understanding of the topic under investigation.

To ensure methodological rigor, specific standards were followed for each study’s component. For the quantitative component, we followed the Strengthening the Reporting of Observational Studies in Epidemiology (STROBE) guidelines [11]. For the qualitative component, the Consolidated Criteria for Reporting Qualitative Research (COREQ) was followed [12]. In addition, the Mixed Methods Appraisal Tool (MMAT) was used to ensure the appropriateness of the study report [13].

### 2.2. Ethical Considerations

The study was conducted in accordance with the Declaration of Helsinki, as per Resolution No. 466/2012 of the National Health Council, which regulates research involving human beings in Brazil. It was approved by the Research Ethics Committee of the Federal University of Espírito Santo, under number 5,519,362, approval date 11 July 2022. Confidentiality regarding the privacy of sensitive data was ensured throughout the study, with participants identified by codes (e.g., P1, P2, or P296). Participants were informed of the purpose and stages of the research and were informed that they could withdraw from the study at any time without any interference with their hospital care and treatment. Interviews were conducted in a private setting after signing two copies of the Free and Informed Consent Form.

### 2.3. Study Setting

The study was conducted in a public maternity ward of a teaching hospital located in the city of Vitória, Espírito Santo state, Brazil. The hospital serves as a regional referral center for medium- and high-complexity care, offering exclusively public services that influence the health of the local population and neighboring states. The maternity ward is a reference in high-risk obstetric care, covering both spontaneous demand from the municipality and cases referred from neighboring cities and more complex referrals. It recorded 970 live births in 2023. According to institutional protocol and Ministry of Health guidelines, newborns with a gestational age ≥ 34 weeks or weight ≥ 2000 g are admitted to rooming-in care, being referred to the neonatal unit only in cases of clinical instability. In the sample analyzed, 57% of the newborns were late preterm (34–36+6 weeks), reflecting the high-risk profile of the population served.

### 2.4. Population and Eligibility Criteria

The study population consisted of postpartum women admitted to the rooming-in unit. This hospital-based practice enables mothers and neonates to remain in the same room throughout their hospitalization, promoting bonding and facilitating breastfeeding.

The inclusion criteria for participants were mothers aged over 18 years, currently breastfeeding, neonates ≥ 34 weeks, a minimum hospital stay of 48 h, and an active telephone number for longitudinal follow-up. Women with contraindications to breastfeeding were considered ineligible.

Loss criteria included withdrawal from follow-up and inability to establish contact after three consecutive attempts on different days and at different times.

### 2.5. Participants

For the quantitative component (QUAN), the sample size was calculated based on the total number of births recorded in the previous year (*n* = 970), using a 5% margin of error and a 95% confidence interval, which indicated a minimum of 276 participants. To compensate for potential sample losses during follow-up, a margin of 7% was added to the minimum estimated sample size, resulting in 296 mothers initially included. This adjustment aimed to ensure the statistical power of the sample in the event of dropouts or anticipated follow-up losses.

After the sample size was determined, qualitative and quantitative data were collected concurrently. The decision to conduct qualitative interviews with all eligible participants, rather than stopping at thematic saturation, stemmed from the methodological strategy adopted in the convergent parallel design [14]. This approach was intended to allow all narratives to be directly compared with the corresponding quantitative variables, thereby strengthening the robustness of the integrated results.

During the follow-up phase of the quantitative component, 22 participants (7.43%) withdrew from the study, resulting in a final quantitative sample of 274 postpartum women.

### 2.6. Data Collection

The postpartum women were approached in person at least 48 h postpartum, according to institutional protocol, considering that hospital discharge only occurs when mother and baby are clinically stable. Because this is a high-risk maternity hospital, hospital stays exceeding 48 h are common due to complications or breastfeeding difficulties. After being invited to participate in the study, the interviews began after receiving guidance on the research and the postpartum woman’s consent to participate.

QUAL data were collected between April 2024 and January 2025 through individual semi-structured interviews conducted at the bedside in a four-bed shared ward or, when available, in a private room. All interviews were audio-recorded and had an average duration of 10 min. To ensure privacy and confidentiality, measures were taken to minimize participant exposure and/or discomfort, such as using a private room when available or pausing the interview when necessary.

QUAN data were collected from April 2024 to April 2025 through telephone follow-up in four waves, with 48 h, on the 14th, 40th, and 90th days after the infant’s birth. These follow-up points were chosen because they represent key phases of the puerperium: immediate adaptation (48 h), initial adaptation (14 days), puerperal review and resumption of activities (40 days), and increased risk of weaning due to return to work (90 days).

The data collection team consisted of the lead researcher and trained undergraduate nursing students, who had completed 10 h of training on the use of the data collection instruments.

### 2.7. Data Collection Instruments

In the QUAL stage, interviews were conducted using a script consisting of three open-ended questions to understand the influence of guidance received during pregnancy and childbirth on breastfeeding:(i)Tell me about the breastfeeding education you received during prenatal care.(ii)Tell me about the breastfeeding education you received in the delivery room.(iii)Tell me about the breastfeeding education you received in the rooming-in unit.

For the QUAN component, a structured electronic questionnaire was used to collect the following study variables:

Dependent variable: type of feeding offered to the infant.

Independent variables: maternal age (in years), marital status, years of education, number of pregnancies, number of prenatal visits, gestational age at birth, mode of birth, milk supply, and social and family support.

Additional variables included the type of feeding offered since birth (exclusive breastfeeding, formula), difficulty breastfeeding (yes, no), use of artificial teats (including bottle, cup), use of formula (yes, no), infant-related problems (yes, no), and pacifier use (yes, no).

A pretest was conducted with ten women to assess the applicability and clarity of the data collection instruments. Since no adjustments were required, these interviews were incorporated into the final sample without introducing bias or duplication in the analysis.

### 2.8. Data Analysis and Processing

For the QUAL data, interviews were fully transcribed manually by the principal investigator, based on a careful review of the recorded audio, and the text was edited to remove speech fillers, diminutives, and any personal information that could identify individuals mentioned in the interview who were not the primary focus of the study (such as professionals). Transcripts were not returned to participants for comments or corrections, as the analysis focused on interpreting the meanings that emerged from the discourse recorded during the interview, respecting the natural structure of the accounts. To ensure the accuracy of the information, the researchers carefully reviewed the transcripts, preserving both the content and the original context of the narratives. Data analysis was performed using Interface de R pour les Analyses Multidimensionnelles de Textes et de Questionnaires (IRaMuTeQ), version 7.2 [15].

Thematic categories were developed using Descending Hierarchical Classification (DHC), a method that allows for in-depth analysis and is widely applied in health research using IRaMuTeQ.

Partial responses to each question were compiled into a single document, known as the textual corpus. To ensure that the software properly recognized each interview as a separate unit, command lines were inserted to delimit individual interviews. Participants were identified using four codifiers: “ppw,” an abbreviation for postpartum women, followed by a number indicating the chronological order of inclusion; “ms” followed by a number representing marital status: “1” for married/common-law and “2” for never married; parity followed by a number indicating parity: “1” for primiparous, “2” for secundiparous, “3” for terciparous, and “4” for multiparous; and birth followed by a number representing mode of birth: “1” for vaginal birth and “2” for cesarean section. Once the text set was prepared, the material was processed in the software, which segmented the textual corpus into categories, allowing for the extraction of relevant narrative excerpts.

QUAN data were entered into Microsoft^®^ Excel^®^ for Microsoft 365 and analyzed using descriptive statistics, including measures of central tendency (mean and median), position (first quartile—Q1 and third quartile—Q3), and dispersion (standard deviation) for continuous variables, as well as absolute and relative frequencies for categorical variables. Fisher’s exact test, Pearson’s chi-square test, and the Kruskal–Wallis test were applied to assess associations. Cross-sectional analyses were conducted at each follow-up point using binomial logistic regression models to estimate the associations between exclusive breastfeeding and sociodemographic and clinical factors, expressed as odds ratios (ORs) and 95% confidence intervals (95% CIs). To assess temporal trends in the variables, the nonparametric Cochran–Mantel–Haenszel test was applied. All statistical analyses were conducted using R software, version 4.3.3.

### 2.9. Integration of Results

After the separate analysis of the QUAL and QUAN data, identifying points of convergence and divergence integrated the results, with equal weight assigned to both approaches, in accordance with the guidelines proposed by Creswell and Clark [14]. Data integration was carried out using the concurrent embedding strategy [16]. To facilitate the visualization of the integrated findings, a Joint Display was created, allowing for the development of meta-inferences.

## 3. Results

Of the participating mothers, 224 (82%) were between 18 and 35 years old, and 193 (70%) were married or living with a partner. Regarding education, 132 (48%) had completed up to 12 years of schooling, and 61 (22%) had completed 16 years. A total of 101 women (37%) were primiparous, while 173 (63%) had had at least one previous pregnancy, with 71 (26%) being multiparous. Most participants attended more than six prenatal visits, with 234 participants (85%).

Late preterm birth was the most common outcome, occurring in 155 participants (57%), and cesarean section was the predominant mode of delivery, with 198 participants (72%). Regarding infant feeding at the end of follow-up, 156 mothers practiced exclusive breastfeeding (57%), 92 used mixed feeding (34%), and 26 relied exclusively on formula (9%). During hospitalization, 25 infants (22%) received formula supplementation, mostly in association with breastfeeding. Lastly, 254 participants reported having social or family support (93%).

The analysis of associations, using Fisher’s exact test and Pearson’s chi-square test, revealed significant findings regarding the number of prenatal visits, mode of delivery, and breast milk production (Table 1).

Although 85% had ≥6 prenatal consultations, the majority reported no breastfeeding counseling, suggesting a mismatch between coverage and content. Education level was not significantly associated with feeding type, while ≥6 consultations showed a higher prevalence of exclusive breastfeeding in the bivariate analysis (*p* = 0.010); however, the qualitative findings indicate that quantity does not substitute for quality.

The qualitative data on postpartum women’s perceptions of breastfeeding guidance during prenatal care converged with the quantitative findings and revealed a lack of breastfeeding education during prenatal visits, as evidenced by the participants’ statements.


*“During prenatal care, I didn’t receive any breastfeeding guidance. They also didn’t ask whether I wanted to breastfeed or how I intended to do it.”*

*(P08)*



*“There wasn’t even time for guidance during prenatal care.”*

*(P34)*



*“At first, no one talked about breastfeeding. I only learned about it at the hospital from the nurses here. No doctor gave me any guidance during prenatal care.”*

*(P51)*



*“I had seven prenatal visits at the high-risk clinic and the specialty center, but I didn’t receive any breastfeeding counseling.”*

*(P76)*


Although the number of prenatal visits showed a statistically significant association with exclusive breastfeeding, the qualitative data revealed that prenatal care providers failed to offer breastfeeding education.

Breast milk production was significantly associated with the type of infant feeding (*p* = 0.015). Mothers who reported increased milk production had a higher prevalence of exclusive breastfeeding (63%), whereas those who reported low production had a lower exclusivity rate (41%) and a greater use of formula (26%).

Regarding delivery room care, more than half of the mothers reported that immediate skin-to-skin contact with their neonates after birth was respected, and most babies remained in their mothers’ arms for 30 min or for one hour or more. After skin-to-skin contact, the neonates were cleaned, weighed, and received routine care.

These quantitative findings align with participants’ reports in the qualitative phase, which highlight the importance attributed to the emotional and institutional support received during the postpartum period. The following excerpts illustrate how this support was perceived as a positive factor in the breastfeeding experience and in caring for the newborn:


*“In the delivery room, they assessed my breasts and taught me how to massage and express colostrum. The obstetric nurse also guided me on how to properly position the baby and adjust the latch so he could take in the whole areola (…)”*

*(P06)*



*“My experience at the hospital was very positive because they explained what to do, and I didn’t have any difficulties. As soon as he was born, I could feel his touch until I was transferred to the room. They explained how to help him latch onto the breast.”*

*(P39)*



*“I received excellent care. The team provided strong support.”*

*(P155)*



*“I was glad to learn new things. I had experiences I’d never had before. I lived through moments I never imagined. My labor was induced, the nurse brought a robot and played music—I danced in the delivery room, and it was wonderful. When she was born, she was placed in my arms. It was amazing. She stayed with me for a long time until we went to the rooming-in unit—I was never separated from her.”*

*(P165)*


On the other hand, many neonates born via cesarean section did not experience skin-to-skin contact or any other close contact with their babies within the first hour postpartum. In most cases, the absence of skin-to-skin contact was attributed to neonatal complications, such as the need for resuscitation. The interviews revealed women’s perceptions of the conditions of their neonates’ birth and the care provided in the delivery room:


*“The experience was a bit frightening. The anesthesia felt very different. The doctor didn’t bring the baby for breastfeeding right away—she went straight to the NICU. One of the twins was brought to the room, and the other stayed in the NICU because he was premature.”*

*(P47)*



*“When he was born, I saw him, but they didn’t place him on my chest. They said he needed an oxygen mask to help him breathe better. After that, I waited downstairs for the anesthesia to wear off, and when it did, they brought the baby to me.”*

*(P104)*



*“She was born and placed on oxygen. Later, she was brought to stay with me, but she didn’t latch.”*

*(P158)*



*“He was born in the delivery room and didn’t breathe. I didn’t have any contact because they immediately took the baby.“*

*(P174)*



*Other aspects, such as the operating room temperature, were also decisive in preventing skin-to-skin contact.*



*“I didn’t have the golden hour. He was born with a low body temperature, so he stayed with me for only 5 min.”*

*(P212)*



*“I was very nervous. My delivery was urgent because my placenta was detaching, so I had a C-section. I didn’t experience the golden hour because my baby was born with very low body temperature and was immediately taken to the NICU.”*

*(P225)*



*“[…] The baby didn’t stay on my chest for long because he was born with low temperature and needed to be warmed up.”*

*(P285)*


Formula use during hospitalization was significantly associated with feeding type (*p* < 0.001). None of the newborns who were exclusively breastfed had received formula, whereas 22% of those who were given infant formula during their hospital stay maintained that feeding pattern after discharge.

The analysis of variables across the four observation points (48 h, 14 days, 40 days, and 90 days) revealed a significant decline in the proportion of mothers reporting breastfeeding difficulties, from 49% to 5.7% (*p* < 0.001). As these difficulties decreased, an improvement in the prevalence of exclusive breastfeeding was observed, accompanied by a significant reduction in formula use over time (45%, 32%, and 23%, respectively) and a marked variation in pacifier use, with a peak at 14 days (41%) followed by declines at 40 days (32%) and 90 days (18%), both statistically significant (*p* < 0.001).

Other sociodemographic and obstetric characteristics showed no statistically significant differences in feeding type across the analyzed periods (*p* > 0.9). Social support was more commonly reported among mothers who practiced exclusive breastfeeding (93%).

No statistically significant differences were observed between the “mixed feeding” and “no mixed feeding” groups regarding the duration of breastfeeding until discontinuation (median 35 days [Q1 = 14; Q3 = 72] in the formula-exclusive group vs. 32 days [Q1 = 12; Q3 = 63] in the non-formula group; *p* = 0.5), as shown in Table 2. On the other hand, a statistically significant association was observed between formula use during hospitalization and feeding type at the time of analysis (*p* = 0.002).

The analysis of odds ratios for exclusive breastfeeding—using mixed feeding as the comparison group—showed that formula use within the first 48 h of life was significantly associated with reduced odds of exclusive breastfeeding. Neonates who received formula within the first 48 h were five times more likely to be on mixed feeding at hospital discharge than those who were exclusively breastfed (OR = 5.83; 95% CI: 4.04, 8.41; *p* < 0.001). The confidence interval did not include the null value (OR = 1), reinforcing the statistical significance of this association (Table 3).

Neonates who were exclusively formula-fed during the first 48 h of life were 28 times more likely to be formula-fed at discharge rather than exclusively breastfed, compared to those who did not receive formula during hospitalization (OR = 27.7; 95% CI: 11.3–67.8; *p* < 0.001).

A statistically significant association was also identified between the number of days until exclusive breastfeeding cessation and the type of feeding at discharge. The analysis suggests that for each additional day of mixed feeding or formula use (OR = 0.98; 95% CI: 0.97–0.98; *p* < 0.001), the likelihood of maintaining exclusive breastfeeding slightly decreased.

Binomial logistic regression analysis demonstrated that not receiving formula during hospitalization unit was significantly associated with greater odds of exclusive breastfeeding at all follow-up points, with increasing odds ratios: at 14 days, OR = 1.47 (95% CI: 0.74–2.91; *p* < 0.001); at 40 days, OR = 2.52 (95% CI: 1.42–4.51; *p* = 0.007); and at 90 days, OR = 2.15 (95% CI: 1.23–3.80; *p* = 0.007).

The absence of breastfeeding difficulties was associated with higher odds of exclusive breastfeeding, with statistical significance observed only at the 14-day follow-up (OR = 2.56; 95% CI: 1.10–5.80; *p* = 0.030). At this point, the likelihood of exclusive breastfeeding was approximately three times higher when no difficulties were reported, although no significant differences were reported at a later point.

Not using a pacifier was associated with higher odds of exclusive breastfeeding at 40 days (OR = 2.76; 95% CI: 1.56–4.96; *p* < 0.001) and 90 days (OR = 4.03; 95% CI: 2.30–7.18; *p* < 0.001). The odds of exclusive breastfeeding were approximately three times higher at 40 days and between three and nine times higher at 90 days when pacifiers were not used (Table 4).

### Data Integration

The integration of quantitative and qualitative data, available in Table 5, revealed important complementarities and gaps between what is measured objectively and what women experience during the pregnancy–postpartum cycle. The qualitative data broadened the interpretation of the results, enabling the identification of nuances that cannot be captured through numbers, underscoring the importance of considering maternal experience as a legitimate and necessary analytical dimension in evaluating policies and care practices.

## 4. Discussion

The integrated analysis of quantitative and qualitative data revealed that, while there is an association between prenatal care adherence, mode of delivery, breast milk production, and the use of formula during hospitalization, institutional factors continue to negatively affect breastfeeding practices.

The findings point to limitations in institutional strategies to promote breastfeeding, suggesting that guidance provided during hospitalization may be insufficient to counteract the effects of interventionist clinical practices, especially within the first days of life. In this context, the importance of providing effective and individualized support immediately after birth stands out, with a focus on the early management of breastfeeding challenges, such as breast milk production. Supporting this perspective, Flaherman et al. [17] demonstrated that the introduction of formula during the first week of life is strongly associated with early breastfeeding discontinuation, even when hospital counseling is provided.

Additionally, the absence of prenatal counseling and the influence of hospital routines underscore the importance of continuity of care and consistent educational actions beginning in the gestational period. These finding highlights prenatal care not only as a tool for obstetric monitoring but also as a key opportunity for health education and preparation for breastfeeding, particularly in maternity hospitals that serve as referral centers for high-risk pregnancies [18].

This study showed that the relationship between mode of delivery and early infant feeding practices remains a sensitive issue in breastfeeding promotion policies, with evidence suggesting that vaginal delivery supports exclusive breastfeeding, whereas cesarean section is associated with the early use of formula. Other studies support this scenario, indicating that cesarean births and low birth weight are linked to suboptimal breastfeeding practices [19,20]. These findings reinforce the need to reassess hospital protocols, especially in surgical obstetric environments, to create more favorable conditions for breastfeeding starting immediately after delivery.

Another finding of this study is that, despite high attendance at prenatal care, the results indicate weaknesses in the approach to breastfeeding during this period, which contrasts with the expectation of comprehensive care. Pinho-Pompeu et al. [21] also highlighted this issue. They reported that 64.4% of pregnant women did not receive breastfeeding guidance during pregnancy. Such evidence underscores the need to reconsider hospital protocols, especially in surgical obstetric settings, to create more favorable conditions for breastfeeding in the immediate postpartum period.

Therefore, investing in public policies and care practices that strengthen prenatal counseling and provide adequate support for mothers—particularly those in vulnerable situations—is essential to improve exclusive and prolonged breastfeeding rates [22].

Failures in breastfeeding education may be compounded by several factors identified in the literature. Hoyos-Loya et al. [23] highlight a shortage of qualified breastfeeding professionals in healthcare facilities, a lack of culturally appropriate educational materials, gaps in continuity of care, and inadequate guidance on formula use. Added to this is the limited training of physicians and other health professionals in breastfeeding medicine, as discussed by Blitman et al. [24], which contributes to insufficient and often ineffective management when mothers face breastfeeding challenges.

The introduction of formula within the first 48 h of life is strongly associated with the discontinuation of exclusive breastfeeding, leading to mixed or formula feeding patterns by hospital discharge and undermining breastfeeding continuity. These results are consistent with the international literature, which has long linked early, non-medically indicated formula use to early weaning and the interruption of exclusive breastfeeding [25]. Even when the overall duration of breastfeeding is similar between groups, its quality is clearly compromised when formula is introduced during the first moments of the neonate’s life.

From a quantitative perspective, most participants had completed high school. As highlighted by Li et al. [26], women with higher education levels and more favorable socioeconomic conditions are more likely to initiate breastfeeding early and continue it for longer periods. Thus, qualitative analysis offers interpretive support for the statistical findings by showing that these structural conditions directly influence women’s experiences and decisions regarding breastfeeding.

Especially in the context of a high-risk maternity setting, the presence or absence of immediate skin-to-skin contact emerged as a significant moderating factor for these outcomes. Although the study identified that skin-to-skin contact occurred in a portion of births, many mothers reported that it was absent among preterm neonates or those requiring intensive neonatal care immediately after delivery. These situations, which are common in high-risk contexts, compromise the implementation of the so-called “golden hour”—a crucial period for stimulating milk production, initiating early suckling, and strengthening emotional bonding [27]. In the absence of this important practice, perceived social support becomes a fundamental pillar for the success of breastfeeding. Therefore, hospital and educational interventions should be strengthened to mitigate the negative consequences of separation and difficulty initiating lactation [28].

The qualitative findings reinforce this perception by revealing that, although mothers wished for and recognized the benefits of having the baby with them within the first hours after birth, institutional practices often failed to guarantee this right, especially in deliveries involving complications. This finding is consistent with those of Agudelo et al. [29], who demonstrated that complex neonatal adaptations may act as barriers to immediate contact. However, even if not immediately, reestablishing contact as soon as possible can still significantly contribute to exclusive breastfeeding rates and enhance the emotional experience of mothers, as evidenced by Kamrani et al. [30].

Hospital practices, particularly the timing and opportunity for skin-to-skin contact, exert a decisive and measurable influence on breastfeeding outcomes, especially in the initial postpartum period [31]. In contrast, indiscriminate formula use and the absence of skin-to-skin contact have a determinant and measurable impact on the continuation of exclusive breastfeeding. These effects are not always evident in more obvious variables, such as total breastfeeding duration, but they become clear when analyzing the quality and exclusivity of neonatal feeding in the first weeks of life. Early and effective interventions in the neonate’s initial days are crucial to the success of exclusive breastfeeding [32].

However, while the quantitative data revealed negative associations with formula use, the qualitative findings highlighted improvements in breastfeeding support during hospitalization. All mothers reported receiving breastfeeding guidance in the hospital setting, a contrast to what was observed during prenatal care.

The literature indicates that structured educational interventions, particularly those supported by technological resources and participatory methods, are effective in promoting adherence to exclusive breastfeeding, even in more complex clinical contexts [33].

Nevertheless, the positive effects of hospital-based support were not sufficient to offset the negative impacts of early formula introduction, raising important questions about the timing and intensity of educational interventions. In other words, although mothers received proper guidance during hospitalization, the early use of formula (often driven by clinical routines or maternal insecurity in the immediate postpartum period) had already altered the newborn’s feeding pattern.

There is a clear need for integrated strategies that include continuous health education and workforce training to ensure proper implementation of skin-to-skin contact and to emphasize the value of early initiation of breastfeeding, thereby reducing clinical practices that undermine exclusive breastfeeding. This perspective highlights the strategic role of the hospital for the promotion, protection, and support of breastfeeding [31].

Additionally, the importance of hospital-based practices that promote breastfeeding within the first hours of life must be emphasized, with a focus on avoiding the unnecessary introduction of formula, except in clinically justified cases. There is an urgent need for stricter protocols on formula use in maternity hospitals, particularly in high-risk settings where intensified breastfeeding support is essential to prevent adverse short- and long-term outcomes.

The results reinforce the urgency of public policies that prioritize workforce training and the reorganization of perinatal care as essential strategies to promote exclusive and equitable breastfeeding.

### Strengths and Limitations of the Study

Among the study’s strengths, it was conducted in a high-risk public maternity hospital, which provides representation of pregnant women and newborns in more complex conditions, including a high proportion of late preterm infants, who are often underrepresented in breastfeeding research. The prospective cohort design, with follow-up up to 90 days and a mixed approach (quantitative and qualitative), allowed for the identification of both relevant statistical associations and maternal perceptions of the breastfeeding process.

On the other hand, some limitations should be considered. The inclusion criterion of a minimum stay of 48 h may have generated selection bias by underrepresenting cases of early discharge, although rare in high-risk settings, which are potentially associated with a higher prevalence of EBF. Furthermore, important variables were not systematically measured, such as: (i) initiation of breastfeeding within the first hour of life/skin-to-skin contact; (ii) number of breastfeeding support visits during hospitalization, as well as a prioritization score; (iii) use of expressed breast milk and access to hospital or private pumps; and (iv) provision of formal post-discharge support services. Although qualitative reports have demonstrated massage and manual expression practices in the immediate postpartum period, the lack of systematic recording of these aspects limits the robustness of the analyses.

Furthermore, the high rate of cesarean sections reflects the institution’s care profile but limits comparability with low-risk populations. The 90-day follow-up, while relevant, does not allow for the assessment of the maintenance of exclusive breastfeeding for the recommended six months. Finally, the use of maternal reports is subject to recall bias. These aspects should be considered when interpreting the findings and guide recommendations for future research.

## 5. Conclusions

This study demonstrated that breastfeeding type changed over time depending on factors such as the number of prenatal visits, mode of delivery, and, most significantly, formula use within the first 48 h of life. Early formula use was found to be the most influential factor in shortening the duration of exclusive breastfeeding.

Higher educational attainment and a greater number of prenatal visits were associated with better breastfeeding practices; however, the quality of counseling received proved to be a decisive factor. The gap in prenatal education on breastfeeding and the early introduction of formula represented major obstacles, particularly in contexts of social vulnerability.

The study identified specific factors that influence exclusive breastfeeding in our sample, such as sociodemographic characteristics, social and institutional support, and local public health practices. These findings are essential for informing public policies and intervention strategies tailored to the needs of the Brazilian population, contributing to improving exclusive breastfeeding rates in the country.

## Figures and Tables

**Table 1 nutrients-17-02992-t001:** Distribution of feeding type at different postpartum follow-up points according to sociodemographic, obstetric, and neonatal variables.

Variables		Feeding Type				Observation Time Point				
		Exclusive *n* = 156	Mixed *n* = 92	Formula * *n* = 26	*p*-Value	48 h *n* = 156	14 Days *n* = 221	40 Days *n* = 181	90 Days *n* = 135	*p*-Value
Years of education					0.2					>0.9
	9 years	27 (68%)	9 (23%)	4 (10%)		27 (24%)	33 (29%)	31 (27%)	22 (19%)	
	12 years	111 (58%)	64 (33%)	18 (9.3%)		111 (23%)	155 (32%)	128 (26%)	95 (19%)	
	16 years	18 (44%)	19 (46%)	4 (9.8%)		18 (20%)	33 (36%)	22 (24%)	18 (20%)	
Number of pregnancies					>0.9					>0.9
	Primiparous	56 (55%)	35 (35%)	10 (9.9%)		56 (23%)	80 (33%)	62 (26%)	44 (18%)	
	Secundiparous	32 (59%)	17 (31%)	5 (9.3%)		32 (21%)	47 (30%)	41 (26%)	35 (23%)	
	Terciparous	29 (60%)	16 (33%)	3 (6.3%)		29 (22%)	42 (32%)	36 (27%)	26 (20%)	
	Multiparous	39 (55%)	24 (34%)	8 (11%)		39 (24%)	52 (32%)	42 (26%)	30 (18%)	
Number of prenatal visits					0.010					>0.9
	Fewer than 3	2 (33.3%)	2 (33.3%)	2 (33.3%)		2 (15%)	6 (46%)	4 (31%)	1 (7.7%)	
	From 3 to 5	9 (56%)	6 (38%)	1 (6%)		9 (21%)	13 (31%)	12 (29%)	8 (19%)	
	6 or more	145 (58%)	84 (33%)	23 (9%)		145 (23%)	202 (32%)	165 (26%)	126 (20%)	
Gestational age					0.6					0.9
	Late preterm	85 (55%)	53 (34%)	17 (11%)		85 (22%)	124 (32%)	106 (27%)	78 (20%)	
	Full term	71 (60%)	39 (33%)	9 (7.6%)		71 (24%)	97 (32%)	75 (25%)	57 (19%)	
Mode of delivery					0.058					0.8
	Cesarean	104 (53%)	73 (37%)	21 (11%)		104 (22%)	153 (33%)	121 (26%)	86 (19%)	
	Vaginal birth	52 (68%)	19 (25%)	5 (6.6%)		52 (23%)	68 (30%)	60 (26%)	49 (21%)	
Breast milk production					0.015					>0.9
	Low	16 (41%)	13 (33%)	10 (26%)		16 (23%)	25 (35%)	18 (25%)	12 (17%)	
	Normal	96 (58%)	57 (35%)	12 (7.3%)		96 (22%)	138 (32%)	112 (26%)	83 (19%)	
	Increased	44 (63%)	22 (31%)	4 (5.7%)		44 (23%)	58 (30%)	51 (26%)	40 (21%)	
Social or family support					0.2					0.9
	Yes	145 (57%)	87 (34%)	22 (8.7%)		145 (23%)	206 (32%)	167 (26%)	123 (19%)	
	No	11 (55%)	5 (25%)	4 (20%)		11 (21%)	15 (29%)	14 (27%)	12 (23%)	
Formula use during hospitalization **					<0.001					<0.001
	Yes	0 (0%)	89 (78%)	25 (22%)		0 (0%)	88 (45%)	62 (32%)	45 (23%)	
	No	98 (98%)	2 (2.0%)	0 (0%)		156 (31%)	133 (27%)	119 (24%)	90 (18%)	
Breastfeeding difficulties					--					<0.001
	Yes	--	--	--		43 (49%)	25 (28%)	15 (17%)	5 (5.7%)	
	No	--	--	--		105 (18%)	193 (33%)	165 (28%)	130 (22%)	
Pacifier use					--					<0.001
	Yes	--	--	--		20 (8.9%)	92 (41%)	72 (32%)	41 (18%)	
	No	--	--	--		136 (29%)	128 (27%)	109 (23%)	94 (20%)	

Kruskal–Wallis’s rank sum test; Pearson’s Chi-squared test; Fisher’s exact test; * Use of formula during hospitalization; ** Type of feeding at the end of follow-up.

**Table 2 nutrients-17-02992-t002:** Association between formula use during hospitalization, obstetric factors, and breastfeeding type.

Variables		Formula Use During Hospitalization			Feeding Type			
		Yes *n* = 100	No *n* = 105	*p*-Value	Exclusive *n* = 80	Mixed *n* = 98	Formula *n* = 27	*p*-Value
Time until breastfeeding discontinuation (days)				0.5				0.14
	Mean (SD)	51 (52)	44 (42)		55 (51)	43 (44)	42 (48)	
	Median (Q1–Q3)	35 (14–72)	32 (12–63)		44 (19–69)	31 (11–65)	25 (5–67)	
Number of prenatal visits (n, %)				0.3				0.6
	Fewer than 3	9 (69%)	4 (31%)		6 (46%)	5 (38%)	2 (15%)	
	3 to 5	9 (45%)	11 (55%)		9 (45%)	7 (35%)	4 (20%)	
	6 or more	82 (48%)	90 (52%)		65 (38%)	86 (50%)	21 (12%)	
Formula use within the first 48 h (n, %)				--				0.002
	Yes	--	--		29 (29%)	51 (51%)	20 (20%)	
	No	--	--		51 (49%)	47 (45%)	7 (6.7%)	

Pearson’s Chi-squared test.

**Table 3 nutrients-17-02992-t003:** Multinomial logistic regression models for factors associated with breastfeeding type at hospital discharge.

Variables		Mixed			Formula		
		OR ^2^	95% CI	*p*-Value	OR	95% CI	*p*-Value
Breastfeeding days		0.98	0.97, 0.98	<0.001	0.98	0.97, 0.98	<0.001
Prenatal visits				0.74			0.74
	Fewer than 3	—	—		—	—	
	3 to 5	1.79	0.55, 5.86		2.11	0.27, 16.6	
	6 or more	1.91	0.73, 5.00		1.91	0.36, 10.2	
Formula use within the first 48 h				<0.001			<0.001
	No	—	—		—	—	
	Yes	5.83	4.04, 8.41		27.7	11.3, 67.8	

CI = Confidence Interval, OR = Odds Ratio; Multinomial logistic regression models.

**Table 4 nutrients-17-02992-t004:** Binomial logistic regression models for factors associated with exclusive breastfeeding at 14, 40, and 90 days postpartum.

Characteristics		14 Days			40 Days			90 Days		
		OR	95% CI	*p*-Value	OR	95% CI	*p*-Value	OR	95% CI	*p*-Value
Formula use during hospitalization				0.27			<0.001			0.007
	Yes	—	—		—	—	—	—	—	
	No	1.47	0.74, 2.91		2.52	1.42, 4.51		2.15	1.23, 3.80	
Breastfeeding difficulties				0.030			0.31			0.064
	Yes	—	—		—	—	—	—	—	
	No	2.56	1.10, 5.80		1.55	0.66, 3.62		2.66	0.95, 8.77	
Pacifier use				0.062			<0.001			<0.001
	Yes	—	—		—	—	—	—	—	
	No	1.91	0.97, 3.84		2.76	1.56, 4.96		4.03	2.30, 7.18	

Abbreviations: CI = Confidence Interval, OR = Odds Ratio; Model: Binomial logistic regression.

**Table 5 nutrients-17-02992-t005:** Triangulation of findings.

Category	Quantitative	Qualitative	Convergence	Divergence	Synthesis
Prenatal visits	85% had more than six prenatal visits; exclusive breastfeeding at discharge was observed in 58% of these cases (*p* = 0.010).	Category 1: Lack of breastfeeding education during prenatal care, even among women with many visits. Ex.: *“I had seven prenatal visits at the high-risk clinic and the specialty center. But I didn’t receive any breastfeeding counseling.”* (P76)	High prenatal care adherence	Quantity ≠ Quality of prenatal visits; subjective data reveal gaps not captured by formal metrics regarding breastfeeding education.	Although there was a correlation between the number of prenatal visits and exclusive breastfeeding, the quality of prenatal breastfeeding support was limited, as evidenced by reports pointing to gaps in education.
Formula use during hospitalization	Strong association with lower likelihood of exclusive breastfeeding (OR = 5.83; *p* < 0.001); exclusive formula use at discharge (OR = 27.7). No statistically significant difference in breastfeeding duration based on formula use during hospitalization (*p* = 0.5).	Category 2: Introduction of formula due to cesarean sections/neonatal complications. Ex.: *“…And then as soon as they brought her, they offered formula. I don’t know why”* (P24). *“…But when he got to the room, I found out they had already given him formula. Then, when I was trying to breastfeed him, a nursing technician came and said I wouldn’t be able to do it at that moment and took him away to give him formula. I was extremely upset because I didn’t want him to have formula at all.”* (P34)	Confirms that formula use is inversely associated with exclusive breastfeeding	Quantitative data does not show an impact on duration. Qualitative findings indicate that healthcare professionals did not inform women about the reasons for formula use.	Formula use interferes negatively with exclusive breastfeeding. Subjective data reveals nuances not captured by objective metrics.
Professional/Social/Family support	93% reported receiving social or family support, but no statistically significant association was found with exclusive breastfeeding at discharge (*p* > 0.2).	Category 4: Recognition of institutional support by postpartum women. Some mothers emphasized: *“The professionals, the nurses, and the residents explained everything and helped me a lot. Support and having the baby on my chest helped with the latch.”* (P39) The emphasis was on team action as a facilitator of breastfeeding.	Recognition of the role of support	No statistical association was found with exclusive breastfeeding, despite participants’ recognition of the relevance of support.	Support is experienced as important for maternal well-being, although its direct impact on infant feeding outcomes was not statistically confirmed.
Mode of delivery and early contact	Trend toward an association between vaginal birth and exclusive breastfeeding (*p* = 0.058); early skin-to-skin contact was not measured.	Category 3: Vaginal birth with immediate skin-to-skin contact supports exclusive breastfeeding, as illustrated in the following accounts: *“As soon as he was born, within seconds they placed him on my chest, and he latched right away. He stayed on my chest for over an hour. Then they cleaned him up, dressed him, and weighed him.”* (P02) By contrast, cesarean delivery impaired early breastfeeding: *“They didn’t place the baby on my chest; instead, he went straight to oxygen.”* (P101)	Accounts reinforce that immediate skin-to-skin contact supports exclusive breastfeeding.	Quantitative data did not directly measure skin-to-skin contact; mode of delivery was not associated with formula use (*p* = 0.7).	The lack of specific variable limits statistical analysis; qualitative data highlights that early skin-to-skin contact is crucial for exclusive breastfeeding, especially in cesarean birth contexts.

## Data Availability

All data analyzed during this study are included in this published article.

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
