# Peer review of "Factors Influencing Exclusive Breastfeeding During the Postpartum Period: A Mixed-Methods Study"

_nutrients, 2025, doi:10.3390/nu17182992_

Round 1
Reviewer 1 Report
Comments and Suggestions for Authors
Dear Authors,
I have read your article carefully and would like to make the following comments:
Abstract:
-why were days 14, 40 and 90 postpartum chosen for evaluation?
-line 27-29: instead of presenting in the Abstract the type of statistical analysis used, the authors could present data on the factors that were followed and that could influence breastfeeding, factors that are then mentioned in the Conclusions.
-line 33-36: already known information.
-line 34-38 Results: only mentions the influence of vaginal birth and skin to skin contact while in the Conclusions they also mention other factors (number of pregnancies, educational altainment, hospital support).
I believe that the Abstract should be rewritten.
Keywords: I suggest “prenatal counseling” instead of “prenatal care”.
Introduction:
-line-46-47: “breastfeeding type (e.g., exclusive, mixed or formula predominant)”: breast feeding is not similary with the administration of formula milk. The administration of formula milk is not a method of breastfeeding. Correct is feeding type.
Material and Methods
-line 84: the year in which the research was approved by the Ethics Committee is not specified.
-line 102: Inclusion criteria instead of “eligibility”
-line 147 (i.i.): I consider that question cannot be addressed in the delivery room, both in the case of mothers who gave birth vaginally, and those who gave birth by cesarean section, with anesthesia. At those times, I consider that breastfeeding education cannot be done.
-line 145: “breastfeeding type” – authors are asked to specify which type of breastfeeding they were referring to! I believe the word “type” can be omitted.
-in my opinion, the lack of a questionnaire for QUAL with the same questions to be answered by all mothers and the administration of only 3 questions (line 146-148) can lead to error, each mother answering according to what she remembers or depending on her level of education. It should be specified who did educational counseling during pregnancy (family practitioner, gynecologist, nurse) and how many counseling sessions they attended.
-line 156: “water, teas, other” – the administration of water and tea (probably between meals with milk) does not represent a main “type of feeding”.
Results:
-line 212: “16 years” means higher education?
-line 214: I think the correct answer is: of which 71 being multiparous (4 or more children).
-line 216-217: I don’t think it’s correct to mention both the number and the percentage in the same parenthesis.
-line 218: “26 (9%) exclusively on formula” – maybe the cases with exclusive formula feeding should be excluded, considering the title of the article. Regarding the cases that received mixed feeding (human milk + formula) I suggest presenting the data in comparison with those with exclusive breastfeeding.
The results mentioned in lines 210-219 are redundant with those in table 1.
-Table 1- formula use during hospitalization: I think the numbers are wrong: in table 25 (22%) on formula, then in line 3 of table- 26, as well as in line 216: 26 (9%) on formula.
-Table 1: “breastfeeding difficulties”: the number of cases that had problems or not at certain times (48 hours, 14/40/90 days) is mentioned, but without making correlations with feeding type (e.g., how many of those with exclusive breastfeeding presented difficulties at 48 hours, 14/40/90 days). Same observation for “pacifier use”.
-although in the Abstract the authors mentioned that days 14, 40 and 90 post-partum were chosen for evaluation, in table 1 the evaluation at 48 hours is also presented. It is not mentioned in Methods either.
-line 225: the number of prenatal visits is equal to the number of educational counselling visits?
-line 232-235: the results are presented in general terms, without specifying exactly (%) how many received adequate educational counselling on breastfeeding and how many did not from the mothers' point of view.
-line 309-312: when we say “formula group” we mean those fed only formula. I believe that the paragraph should be reworded: instead of “formula group” and “formula used during hospitalization” we should mention “mixed feeding”. Otherwise, we cannot compare the duration of breastfeeding (line 310).
Table 2: number of formula use = 105, but in Table 1: formula use = 92+26 = 118! In the same table 1: “formula using during hospitalization” : 89+25 = 114!
I believe that the statistical analysis should be verified by an experienced statistician.
Discussions: the authors discuss the results obtained in comparison with the results obtained in other studies. They do not present the strengths and weaknesses / limitations of the study.
The authors do not present data on the impact of the mental health of post-partum women on breastfeeding.
References: small number of references given the very large number of articles published over the years, including in the last 5 years.
I believe that, in the context of a large number of articles already published, the article has limited value.
Comments on the Quality of English Language
Proofreading of the English language by a specialized person is necessary.
Author Response
Dear Healthcare Editor-in-Chief,
Thank you for the opportunity to review our manuscript. We have addressed all the issues suggested by the reviewers, which has significantly improved our work. All changes are highlighted in the main text for your convenience.
Abstract:
Reviewer's comment: Why were days 14, 40 and 90 postpartum chosen for evaluation?
Response: I greatly appreciate your valuable observations and suggestions for improving the clarity and consistency of this research. Therefore, we justify the choice of days 14, 40, and 90 postpartum in the methods section. A brief justification is included in the summary section.
The selection of days 14, 40, and 90 postpartum reflects key milestones in establishing and sustaining breastfeeding. Day 14 marks the early phase of mother–infant adaptation, a period often characterized by challenges such as latching difficulties, pain, and the risk of early formula introduction. Day 40 typically aligns with the routine postpartum check-up and the end of the confinement period, when many women begin to resume daily activities, potentially affecting exclusive breastfeeding. Day 90 coincides with the end of Brazil’s minimum maternity leave for some women, making the return to work a critical point of vulnerability for early weaning. Monitoring these three time points enabled the identification of periods of greater risk for the transition away from exclusive breastfeeding, capturing initial, intermediate, and maintenance changes in the infant’s feeding pattern.
Reviewer's comment: Line 27-29: instead of presenting in the Abstract the type of statistical analysis used, the authors could present data on the factors that were followed and that could influence breastfeeding, factors that are then mentioned in the Conclusions.
Response: Thank you for your observation. We agree that the Methods section of the abstract did not clearly specify which factors were monitored throughout the analysis. We have revised the text to explicitly include sociodemographic factors (age, education, marital status, number of pregnancies), clinical factors (gestational age, mode of delivery, milk production), and support-related factors (social and hospital), as well as the use of formula during hospitalization. These adjustments ensure that the factors discussed in the conclusions are now clearly stated and consistently aligned with the Methods section of the abstract.
Reviewer's comment: line 33-36: already known information.
Response: Thank you for your observation. Please note that the wording of lines 33-36 has been revised to improve clarity and accuracy of the information presented.
Reviewer's comment: line 34-38 Results: only mentions the influence of vaginal birth and skin to skin contact while in the Conclusions they also mention other factors (number of pregnancies, educational altainment, hospital support).
Response: Thank you for the pertinent notes. Results rewritten.
Reviewer's comment: I believe that the Abstract should be rewritten.
Response: Done.
Reviewer's comment: Keywords: I suggest “prenatal counseling” instead of “prenatal care”.
Response: We appreciate the suggestion to change the keyword; however, the term "prenatal counseling" is not included in the health sciences descriptors and Mesh terms. Therefore, the keyword was changed to "prenatal education." If the reviewer still considers it appropriate to change it to "prenatal counseling," we will do so.
Introduction:
Reviewer's comment: line-46-47: “breastfeeding type (e.g., exclusive, mixed or formula predominant)”: breast feeding is not similary with the administration of formula milk. The administration of formula milk is not a method of breastfeeding. Correct is feeding type.
Response: I appreciate the pertinent point, and I agree with the change in expression. The text was revised to replace 'type of breastfeeding' with 'type of feeding,' correctly reflecting the distinction between breastfeeding and formula milk administration.
Material e Métodos
Reviewer's comment: line 84: the year in which the research was approved by the Ethics Committee is not specified.
Response: I inform you that the information about the date of approval by the Ethics Committee is described in the Method section, under the item 'Ethical Considerations'."
Ethical Considerations
The study was conducted in accordance with the Declaration of Helsinki, as per Resolution No. 466/2012 of the National Health Council (CNS), which regulates re-search involving human beings in Brazil. It was approved by the Research Ethics Committee (CEP) of the Federal University of Espírito Santo (UFES), under number 5,519,362, approval date 2022-07-11. Confidentiality regarding the privacy of sensitive data was ensured throughout the study, with participants identified by codes (e.g., P1, P2, or P296). Participants were informed of the purpose and stages of the research and were informed that they could withdraw from the study at any time without any interference with their hospital care and treatment. Interviews were conducted in a private setting after signing two copies of the Free and Informed Consent Form (FICF).
Reviewer's comment: line 102: Inclusion criteria instead of “eligibility”
Response: Done.
Reviewer's comment: line 147 (i.i.): I consider that question cannot be addressed in the delivery room, both in the case of mothers who gave birth vaginally, and those who gave birth by cesarean section, with anesthesia. At those times, I consider that breastfeeding education cannot be done.
Response: "Thank you for your observation. We would like to clarify that breastfeeding education is indeed an integral part of the routine care provided in the delivery room after both vaginal and cesarean births, particularly during the “golden hour,” as widely recommended by the World Health Organization (WHO) and the Brazilian Ministry of Health. We therefore consider this a timely and safe opportunity to guide and support mothers in initiating breastfeeding immediately after birth.
Reviewer's comment: line 145: “breastfeeding type” – authors are asked to specify which type of breastfeeding they were referring to! I believe the word “type” can be omitted.
Response: Thank you for your observation. The sentence has been revised, and the expression 'type of breastfeeding' has been replaced with 'breastfeeding,' making the text clearer and more objective. "In the QUAL stage, interviews were conducted using a script consisting of three open-ended questions to understand the influence of guidance received during pregnancy and childbirth on breastfeeding."
Reviewer's comment: in my opinion, the lack of a questionnaire for QUAL with the same questions to be answered by all mothers and the administration of only 3 questions (line 146-148) can lead to error, each mother answering according to what she remembers or depending on her level of education. It should be specified who did educational counseling during pregnancy (family practitioner, gynecologist, nurse) and how many counseling sessions they attended.
Response: I appreciate the comments. I would like to clarify that, in qualitative studies, the use of open-ended questions aims to explore participants' discourses in a flexible manner, allowing emerging themes to be explored in depth during the interview. The three open-ended questions mentioned above served to initiate the discussion, and the interviewer guided and deepened the conversation as the mothers' accounts emerged, which is consistent with the qualitative research approach based on semi-structured interviews.
Regarding who provided educational counseling during the prenatal period, this information was mentioned by the participants in their statements. Some details may have been omitted in excerpts of the statements, out of respect for anonymity, as some mothers even mentioned the names of professionals. Furthermore, the number of consultations and information about prenatal care and breastfeeding were already investigated in the QUAN stage, through the collection of sociodemographic and clinical obstetric data. Therefore, the aspects raised by the reviewer are addressed, respecting the qualitative methodology and the ethical considerations of the study.
Reviewer's comment: line 156: “water, teas, other” – the administration of water and tea (probably between meals with milk) does not represent a main “type of feeding”.
Response: I would like to inform you that the words 'water, teas, and others' have been removed from the text, as suggested, since they do not represent a main type of diet. Thank you.
Results
Reviewer's comment: line 212: “16 years” means higher education?
Response: I confirm that '16 years' refers to higher education, adopting this form of presentation due to the various classifications of education that exist internationally.
Reviewer's comment: line 214: I think the correct answer is: of which 71 being multiparous (4 or more children).
Response: Thank you for your observation. The section about multiparous women has been revised for greater accuracy: of the 173 participants who had already experienced at least one pregnancy, 71 were multiparous (having four or more children).
Reviewer's comment: line 216-217: I don’t think it’s correct to mention both the number and the percentage in the same parentheses.
Response: I would like to inform you that the paragraph has been adjusted, separating absolute numbers from percentages, as suggested, for greater clarity and adherence to data presentation standards.
Reviewer's comment “26 (9%) exclusively on formula” – maybe the cases with exclusive formula feeding should be excluded, considering the title of the article. Regarding the cases that received mixed feeding (human milk + formula) I suggest presenting the data in comparison with those with exclusive breastfeeding.
Response: We understand the reviewer’s suggestion; however, excluding the cases of exclusive formula feeding would impact the analysis of the present study. Conducting a comparative analysis between mixed feeding and exclusive breastfeeding would require performing additional statistical tests, which were beyond the scope of this study. Nevertheless, we acknowledge the relevance of this approach and will consider it for a future analysis.
The results mentioned in lines 210-219 are redundant with those in table 1.
Reviewer's comment: Table 1- formula use during hospitalization: I think the numbers are wrong: in table 25 (22%) on formula, then in line 3 of table- 26, as well as in line 216: 26 (9%) on formula.
Response: Thank you for your observation. I would like to clarify that there is no discrepancy between the numbers presented in the text and in Table 1, as they refer to different data collection periods: the 25 (22%) figure corresponds to formula use during hospitalization, while the 26 (9%) presented in the text refer to exclusive formula use at the end of follow-up. To avoid misinterpretations, we revised the text and the caption to Table 1, making this temporal difference more explicit.
Reviewer's comment: -Table 1: “breastfeeding difficulties”: the number of cases that had problems or not at certain times (48 hours, 14/40/90 days) is mentioned, but without making correlations with feeding type (e.g., how many of those with exclusive breastfeeding presented difficulties at 48 hours, 14/40/90 days). Same observation for “pacifier use”.
Response: Thank you for this valuable observation. We did not perform this specific data analysis correlating breastfeeding difficulties or pacifier use with feeding type at the different time points (48 hours, 14, 40, and 90 days). We will, however, carefully consider the possibility of including such an analysis in future studies or as part of a subsequent evaluation.
Reviewer's comment: although in the Abstract the authors mentioned that days 14, 40 and 90 post-partum were chosen for evaluation, in table 1 the evaluation at 48 hours is also presented. It is not mentioned in Methods either.
Response: The comment was very pertinent, thank you. Indeed, the abstract only mentioned days 14, 40, and 90 postpartum, while Table 1 also included the 48-hour assessment, which was not described in the Methods. I clarify that this was a writing inconsistency, which has since been corrected in the manuscript, so that now all sections include information about the 48-hour assessment, in addition to the other follow-up periods (days 14, 40, and 90).
Reviewer's comment: line 225: the number of prenatal visits is equal to the number of educational counselling visits?
Response: As described in paragraphs 3 and 4 of the Results section, the number of prenatal visits did not necessarily correspond to the amount of educational guidance or counseling on breastfeeding received during these visits. Thus, although an association was identified between a higher number of visits and a higher prevalence of exclusive breastfeeding, the qualitative data revealed that prenatal attendance alone did not guarantee the provision of consistent information on breastfeeding. Therefore, the number of prenatal visits and the quantity/quality of counseling were distinct variables in our study.
Reviewer's comment: line 232-235: the results are presented in general terms, without specifying exactly (%) how many received adequate educational counselling on breastfeeding and how many did not from the mothers' point of view.
Response: Thank you for your comment. I would like to clarify that, as described in the methodology, the qualitative portion of the study was conducted through individual interviews with open-ended questions, allowing for the analysis of the postpartum women's discourse. For this reason, this data was not quantified in percentage terms, but rather analyzed interpretatively and correlated with the quantitative findings, using a mixed approach. This methodological strategy was chosen precisely to capture perceptions, meanings, and experiences regarding breastfeeding guidance during prenatal care, which would not be adequately expressed through numbers.
Reviewer's comment: line 309-312: when we say “formula group” we mean those fed only formula. I believe that the paragraph should be reworded: instead of “formula group” and “formula used during hospitalization” we should mention “mixed feeding”. Otherwise, we cannot compare the duration of breastfeeding (line 310).
Response: Thank you. I've refined the wording to match the note.
No statistically significant differences were observed between the “mixed feeding” and “no mixed feeding” groups regarding the duration of breastfeeding until discontinuation (median 35 days [Q1 = 14; Q3 = 72] in the formula-exclusive group vs. 32 days [Q1 = 12; Q3 = 63] in the non-formula group; p = 0.5), as shown in Table 2.
Reviewer's comment: able 2: number of formula use = 105, but in Table 1: formula use = 92+26 = 118! In the same table 1: “formula using during hospitalization” : 89+25 = 114!
Response: I clarify that, in Table 2, the value of N = 105 refers to the group of postpartum women who did not use formula during their hospitalization (as opposed to the 100 who did). In Table 1, in turn, the numbers presented include different segments of the variable "formula use" (for example, use during hospitalization and use after discharge), which may give the impression of inconsistency.
Reviewer's comment: Discussions: the authors discuss the results obtained in comparison with the results obtained in other studies. They do not present the strengths and weaknesses / limitations of the study.
Response: Done.
Among the study's strengths, it was conducted in a high-risk public maternity hospital, which provides representation of pregnant women and newborns in more complex conditions, including a high proportion of late preterm infants, who are often underrepresented in breastfeeding research. The prospective cohort design, with follow-up up to 90 days and a mixed approach (quantitative and qualitative), allowed for the identification of both relevant statistical associations and maternal perceptions of the breastfeeding process.
On the other hand, some limitations should be considered. The inclusion criterion of a minimum stay of 48 hours may have generated selection bias by underrepresenting cases of early discharge, although rare in high-risk settings, potentially associated with a higher prevalence of EBF. Furthermore, important variables were not systematically measured, such as: (i) initiation of breastfeeding within the first hour of life/skin-to-skin contact; (ii) number of breastfeeding support visits during hospitalization, as well as a prioritization score; (iii) use of expressed breast milk and access to hospital or private pumps; and (iv) provision of formal post-discharge support services. Although qualitative reports have demonstrated massage and manual expression practices in the immediate postpartum period, the lack of systematic recording of these aspects limits the robustness of the analyses.
Furthermore, the high rate of cesarean sections reflects the institution's care profile but limits comparability with low-risk populations. The 90-day follow-up, while relevant, does not allow for the assessment of the maintenance of exclusive breastfeeding for the recommended six months. Finally, the use of maternal reports is subject to recall bias. These aspects should be considered when interpreting the findings and guide recommendations for future research.
Reviewer's comment: The authors do not present data on the impact of the mental health of post-partum women on breastfeeding.
Response: We appreciate this important recommendation and fully recognize the relevance of addressing the impact of postpartum women’s mental health on breastfeeding. However, for the present study, no data were collected that would allow for an analysis or discussion of this topic. We acknowledge its significance and will consider incorporating this aspect in future research.
References: small number of references given the very large number of articles published over the years, including in the last 5 years.
I believe that, in the context of a large number of articles already published, the article has limited value.
Reviewer's comment: Proofreading of the English language by a specialized person is necessary.
Response: We received your decision noting that the manuscript requires improvement in English. To address the issues precisely and efficiently, could you please clarify the following:
- Could you provide 2–5 brief excerpts (sentences or sentence fragments) from the manuscript that illustrate the specific type of language problems you identified (for example: unclear sentence structure, ambiguous terminology, frequent grammatical errors that impede comprehension, or wording that alters the meaning of methods/results)? Concrete examples will allow us to correct the exact issues.
- Did the journal prepare an internal language-assessment report or checklist? If so, please share a copy.
Thank you for your time and for clarifying these points.
Reviewer 2 Report
Comments and Suggestions for Authors
Factors influencing exclusive breastfeeding during the postpartum period: A mixed-methods study, by Greyce Minarini et al.
This study aims to examine rates and variables affecting breast feeding up to 12 weeks after birth, using quantitative and qualitative methods, among 296 mothers who gave births at 34 weeks' gestation and above.
Introduction
The authors addressed the importance of breast milk feeding as the ideal nutrition for newborns. They mentioned mixed or formula predominant nutrition when supplemental feeding is required. However, it will be more accurate to state that The World Health Organization (WHO) recommend exclusive breastfeeding for the first 6 months of age for all infants, yet, if mother’s milk is unavailable or insufficient, both preterm and healthy term infants should be fed donor milk. Is fair to say that the use of donor milk in populations other than very low birthweight (VLBW) infants, however, has not been systematically investigated.
Please, can you comment on this in your setting?
Moreover, several high-income countries with high breastfeeding initiation rates, including Australia, Canada, Germany, Norway, and New Zealand, have reported declines in breastfeeding rates at three and four months after birth. Other countries (e.g., the United Kingdom, the Republic of Ireland, the United States, and France) report even lower rates of breastfeeding at the 3–4-month time period, but this is usually coupled with lower initiation rates. This common trend is concerning, in light of the growing evidence regarding the importance of breastfeeding to maternal and child health. Please add.
Methods
Both qualitative and quantitative approaches were used. For the study questions I think that it empowers the results.
I understand that the study was performed among women that were hospitalized in a rooming-in department with their infants. Can the authors explain about the special population of late preterm neonates (34-36 weeks)? What is the percentage? Where do these infants are monitored?
Women were approached 48 hours after birth. Is this the hospital local policy? No earlier discharge? This might cause a bias, as early discharge usually goes with higher lactation rates.
Is expressed breast milk is an option? Do mothers receive information on it? Is equipment available to afford it?
Results
Data and results are well organized and presented.
The authors stated that 85% of the women had at least 6 prenatal visits. Is breast feeding and lactation after birth is a formal part of the prenatal counselling? Can you comment on maternal education and the number of prenatal visits?
Is breast feeding is encouraged immediately after birth? What was the percentage?
How many lactation counselling meetings during hospitalization are routinely conducted? Is there any prioritizing score?
The number of CS is high. Can you comment on it? Can you provide data regarding gestational age and CS rate?
Can the authors comment on lactation support post discharge?
Discussion
Clearly written and covers most important points raised in the results.
Can you comment on cow milk allergy events in cases of mixed formula and breast feeding use?
Comments on the Quality of English Language
No comments.
Author Response
Dear Healthcare Editor-in-Chief,
Thank you for the opportunity to review our manuscript. We have addressed all the issues suggested by the reviewers, which has significantly improved our work. All changes are highlighted in the main text for your convenience.
Introduction:
Reviewer's comment: The authors addressed the importance of breast milk feeding as the ideal nutrition for newborns. They mentioned mixed or formula predominant nutrition when supplemental feeding is required. However, it will be more accurate to state that The World Health Organization (WHO) recommend exclusive breastfeeding for the first 6 months of age for all infants, yet, if mother’s milk is unavailable or insufficient, both preterm and healthy term infants should be fed donor milk. Is fair to say that the use of donor milk in populations other than very low birthweight (VLBW) infants, however, has not been systematically investigated.
Response: Thank you for your comment. I would like to inform you that the opening paragraph of the introduction has been revised in accordance with your recommendations, incorporating WHO guidelines on exclusive and supplementary breastfeeding. I remain available to make any additional adjustments that may be necessary.
Reviewer's comment: Please, can you comment on this in your setting?
Moreover, several high-income countries with high breastfeeding initiation rates, including Australia, Canada, Germany, Norway, and New Zealand, have reported declines in breastfeeding rates at three and four months after birth. Other countries (e.g., the United Kingdom, the Republic of Ireland, the United States, and France) report even lower rates of breastfeeding at the 3–4-month time period, but this is usually coupled with lower initiation rates. This common trend is concerning, in light of the growing evidence regarding the importance of breastfeeding to maternal and child health. Please add.
Methods
Reviewer's comment: Both qualitative and quantitative approaches were used. For the study questions I think that it empowers the results.
I understand that the study was performed among women that were hospitalized in a rooming-in department with their infants. Can the authors explain about the special population of late preterm neonates (34-36 weeks)? What is the percentage? Where do these infants are monitored?
Response: Thank you for your observation. The study was conducted at a high-risk public maternity hospital located in the state capital (Espírito Santo), which serves as an open door to the municipality and surrounding cities, in addition to being a referral center for more complex cases. This care profile frequently involves high-risk pregnancies and, consequently, outcomes such as prematurity.
According to the institutional protocol, in line with the Ministry of Health's recommendations, newborns from 34 weeks of gestational age or weighing ≥2,000 g are admitted to rooming-in care, provided they are clinically stable. NICU admission occurs only in cases with a specific clinical indication.
In the sample, late preterm infants (34–36+6 weeks) represented 57% of the newborns included, reinforcing the high-risk profile of the study population.
Reviewer's comment: Women were approached 48 hours after birth. Is this the hospital local policy? No earlier discharge? This might cause a bias, as early discharge usually goes with higher lactation rates.
Response: Thank you for your valuable considerations. The institution's protocol prioritizes hospital discharge within 48 hours after birth, when mother and baby are healthy, for both vaginal and cesarean deliveries. However, because this is a high-risk maternity hospital, it is more common for babies to remain hospitalized for more than 48 hours, whether due to maternal or neonatal clinical complications or breastfeeding-related difficulties.
Therefore, the decision to begin data collection 48 hours after birth aimed to standardize the inclusion point and homogeneously capture cases treated in rooming-in care. We recognize, however, that this choice may introduce selection bias by underrepresenting babies discharged early, which is rare in this context but could be associated with higher rates of exclusive breastfeeding. This aspect was included in the study limitations.
Reviewer's comment: Is expressed breast milk is an option? Do mothers receive information on it? Is equipment available to afford it?
Response: The use of LME/pumps was not measured as a specific variable. Qualitative reports indicated guidance on massage and manual expression in the immediate postpartum period; however, the availability of pumps (hospital/private) was not systematically mapped.
Results
Reviewer's comment: Data and results are well organized and presented.
Reviewer's comment: The authors stated that 85% of the women had at least 6 prenatal visits. Is breast feeding and lactation after birth is a formal part of the prenatal counselling? Can you comment on maternal education and the number of prenatal visits?
I adjusted the text in the results section to meet the reviewer's suggestion:
Although 85% had ≥6 prenatal consultations, the majority reported no breastfeeding counseling, suggesting a mismatch between coverage and content. Education level was not significantly associated with feeding type, while ≥6 consultations showed a higher prevalence of exclusive breastfeeding in the bivariate analysis (p=0.010); however, the qualitative findings indicate that quantity does not substitute for quality.
Reviewer's comment: Is breast feeding is encouraged immediately after birth? What was the percentage?
Response: There was no specific quantitative variable for “first breastfeeding/SSC in the 1st hour”; qualitatively, more than half reported immediate skin-to-skin contact, especially after vaginal delivery; after cesarean section, the absence of SSC was frequent due to the newborn's clinical conditions. This gap was included as a limitation and improvement roadmap for future collection.
Reviewer's comment: How many lactation counselling meetings during hospitalization are routinely conducted? Is there any prioritizing score?
Response: The number of breastfeeding support interactions per postpartum woman was not measured, nor was a prioritization score used. Qualitatively, postpartum women described frequent and effective support from the team, without quantitative standardization. It is suggested that these indicators be included in future versions.
Reviewer's comment: The number of CS is high. Can you comment on it? Can you provide data regarding gestational age and CS rate?
Response: Thank you for your input. The study was conducted at a high-risk public maternity hospital located in the state capital, which handles the most complex cases in the region. This care profile directly contributes to the high cesarean section rate, as many pregnancies have clinical or obstetric indications for surgical resolution.
In the sample analyzed, the cesarean section rate was 72%, reflecting the increased risk context of the population. We also observed that 57% of the newborns were late preterm (34–36+6 weeks), a group for which the abdominal route is often indicated due to associated maternal or fetal conditions.
Reviewer's comment: Can the authors comment on lactation support post discharge?
Response: Telephone follow-up was carried out (14/40/90 days) for guidance and collection of outcomes, but the post-discharge in-person support network was not mapped. It was observed that not using formula during hospitalization and avoiding pacifiers were associated with higher chances of EBF at 40/90 days (OR=2.52 and OR=2.15; and OR=2.76/4.03, respectively).
Discussion
Reviewer's comment: Clearly written and covers most important points raised in the results.
Can you comment on cow milk allergy events in cases of mixed formula and breast feeding use?
Response: We appreciate the importance of this topic; however, it is beyond the scope of the present study, as we did not investigate issues related to cow’s milk allergy. Therefore, we are unable to provide an analysis or discussion on this aspect. We acknowledge its relevance and suggest it as an important area for future research.
Reviewer 3 Report
Comments and Suggestions for Authors
This study examined factors influencing exclusive breastfeeding during the postpartum period in a Brazilian sample. This is an important topic, and globally there is general improvement in exclusive breastfeeding rates over the past decade, largely because factors have been identified and addressed by healthcare such as the Baby-Friendly Hospital Initiative and the rooming-in initiative. This is a mixed-methods investigation encompassing 274 new mothers over several years. As might be anticipated, the results align with previous research on similar topics (e.g., factors that influence exclusive breastfeeding). There are many meta-analyses reported on this topic – a quick search uncovered 23 meta-analyses on the topic of exclusive breastfeeding and cesarean section alone.
What is the research ‘gap’ the investigators identified to design this research? What new information does this research contribute to existing literature? The results are what would have been predicted based on the wealth of data available on this topic.
The introduction is short and only cites 8 papers. The topic should be reviewed more thoroughly with detail and a comprehensive discussion. Given the vast amount of research on this topic, the data available, and improvements in healthcare delivery targeting exclusive breastfeeding promotion, this short introduction does not accurately portray the current knowledge base in the field. The investigators do not provide justification for this research. Note that mixed-methods are commonly employed in this area of research.
Lines 84-89 etc. acronyms are not needed if not referenced more than 1-2 times in the paper.
The sample size calculation does not include effect size (variability). Please provide more specific detail on this calculation.
Line 119 – please reference this adopted ‘strategy’
Line 212 – ‘maybe 13-16’ seems inappropriate. This should be a specific number determined by the data collected.
Figure 1 should move to the supplemental materials.
Line 390 (also line 437) – based on the comments from the mothers who required cesarean section, it seems there were immediate concerns regarding the newborn’s physical condition that impacted the immediate post-delivery time spent with the newborn. This should be discussed – also discussed should be the many initiatives that have already been enacted in the recent decades to improve on this concern.
Author Response
Dear Healthcare Editor-in-Chief,
Thank you for the opportunity to review our manuscript. We have addressed all the issues suggested by the reviewers, which has significantly improved our work. All changes are highlighted in the main text for your convenience.
This study examined factors influencing exclusive breastfeeding during the postpartum period in a Brazilian sample. This is an important topic, and globally there is general improvement in exclusive breastfeeding rates over the past decade, largely because factors have been identified and addressed by healthcare such as the Baby-Friendly Hospital Initiative and the rooming-in initiative. This is a mixed-methods investigation encompassing 274 new mothers over several years. As might be anticipated, the results align with previous research on similar topics (e.g., factors that influence exclusive breastfeeding). There are many meta-analyses reported on this topic – a quick search uncovered 23 meta-analyses on the topic of exclusive breastfeeding and cesarean section alone.
Reviewer's comment: What is the research ‘gap’ the investigators identified to design this research? What new information does this research contribute to existing literature? The results are what would have been predicted based on the wealth of data available on this topic.
The introduction is short and only cites 8 papers. The topic should be reviewed more thoroughly with detail and a comprehensive discussion. Given the vast amount of research on this topic, the data available, and improvements in healthcare delivery targeting exclusive breastfeeding promotion, this short introduction does not accurately portray the current knowledge base in the field. The investigators do not provide justification for this research. Note that mixed-methods are commonly employed in this area of research.
Response: We appreciate the reviewer's comments and recognize the importance of situating our study within the current context of the scientific literature. We agree that there are indeed several meta-analyses that address factors influencing exclusive breastfeeding, including cesarean section. However, our research stands out for some specific aspects that justify its implementation and contribution to the field.
Reviewer's comment: Lines 84-89 etc. acronyms are not needed if not referenced more than 1-2 times in the paper.
Response: I appreciate the note. done
Reviewer's comment: The sample size calculation does not include effect size (variability). Please provide more specific detail on this calculation.
Response: The variability calculation was not presented separately because it was already considered in the sample sizing, by adopting the maximum uncertainty proportion (p = 0.5; q = 0.5) with a 5% margin of error and a 95% confidence interval. This procedure ensures the highest possible degree of variability and, therefore, the robustness of the final sample of 296 participants.
Reviewer's comment: Line 119 – please reference this adopted ‘strategy’
Response: Done.
Reviewer's comment: Line 212 – ‘maybe 13-16’ seems inappropriate. This should be a specific number determined by the data collected.
Response: Done. I inform you that a typing error occurred, but that the text was corrected in time.
Reviewer's comment: Figure 1 should move to the supplemental materials.
Response: We appreciate the reviewer’s suggestion. We are willing to move Figure 1 to the supplemental materials; however, we would value the Editor-in-Chief’s guidance regarding the best approach to presenting these data and their potential impact on the journal.
Reviewer's comment: Line 390 (also line 437) – based on the comments from the mothers who required cesarean section, it seems there were immediate concerns regarding the newborn’s physical condition that impacted the immediate post-delivery time spent with the newborn. This should be discussed – also discussed should be the many initiatives that have already been enacted in the recent decades to improve on this concern.
Response: We thank the reviewer for this observation. We would like to clarify that this discussion has already been addressed in the Discussion section, where we considered mothers’ reports regarding cesarean delivery and the impact on immediate post-delivery contact with their newborns. Additionally, we have referred to initiatives implemented in recent decades aimed at improving this aspect of care.
Round 2
Reviewer 1 Report
Comments and Suggestions for Authors
Dear authors,
thank you for your answers to my questions and comments. The correction and completion of the article with the recommendations of the first review led to an improvement in the quality of the article.
As for the correctness of the English language, in my opinion, all articles that are not written by native English speakers should be checked by a specialist. An example of less correct expression is the one in lines 213-214 (initial version): “A total of 101 women (37%) were primiparous, while 173 (63%) had had at least 213 one previous pregnancy, with 71 (26%) being multiparous”.